# Migratory functionalization of unactivated alkyl bromides for construction of all-carbon quaternary centers via transposed *tert*-C-radicals

Chuan Zhu [1,3], Ze-Yao Liu[1,3], Luning Tang[1], Heng Zhang[1], Yu-Feng Zhang[1], Patrick J. Walsh [2✉] & Chao Feng [1✉]

Despite remarkable recent advances in transition-metal-catalyzed C(sp³)−C cross-coupling reactions, there remain challenging bond formations. One class of such reactions include the formation of *tertiary*-C(sp³)−C bonds, presumably due to unfavorable steric interactions and competing isomerizations of tertiary alkyl metal intermediates. Reported herein is a Ni-catalyzed migratory 3,3-difluoroallylation of unactivated alkyl bromides at remote tertiary centers. This approach enables the facile construction of otherwise difficult to prepare all-carbon quaternary centers. Key to the success of this transformation is an unusual remote functionalization via chain walking to the most sterically hindered tertiary C(sp³) center of the substrate. Preliminary mechanistic and radical trapping studies with primary alkyl bromides suggest a unique mode of tertiary C-radical generation through chain-walking followed by Ni–C bond homolysis. This strategy is complementary to the existing coupling protocols with *tert*-alkyl organometallic or -alkyl halide reagents, and it enables the expedient formation of quaternary centers from easily available starting materials.

[1] Technical Institute of Fluorochemistry (TIF), Institute of Advanced Synthesis (IAS), School of Chemistry and Molecular Engineering, Nanjing Tech University, 30 South Puzhu Road, 211816 Nanjing, P. R. China. [2] Roy and Diana Vagelos Laboratories, Department of Chemistry, University of Pennsylvania, 231 South 34th Street, Philadelphia, PA 19104, USA. [3]These authors contributed equally: Chuan Zhu, Ze-Yao Liu. ✉email: pwalsh@sas.upenn.edu; iamcfeng@njtech.edu.cn

T ransition-metal-catalyzed construction of all-carbon qua-ternary centers via *tert*-C(sp³)−C coupling reactions repre-sents a significant synthetic challenge. Not only are severe steric effects encountered around the metal center in such coupling reactions, but competing isomerization pathways of alkylmetal intermediates often have low barriers[1–4]. Nevertheless, the devel-opment of cross-coupling protocols that make use of tertiary alkyl −M (M = Mg[5–9], Zn[10], B[11], and Na[12,13]) reagents has aroused substantial interest from the synthetic community. While some advances have been achieved, restricted substrate scopes, together with the need to synthesize the organometallic reagents, has severely limited application of this strategy (Fig. 1a). In this regard, the direct functionalization of *tert*-alkyl electrophiles offers advantages from the perspective of practicality and step economy[14–17]. Of note, recent progress in the area of reductive coupling[18–20] with organic halides and pseudohalides[21–25] or alkenes[26,27] have expanded classes of viable coupling partners. These studies pro-vide complementary and efficient avenues to access structurally diverse three-dimensional scaffolds under mild reaction condi-tions (Fig. 1b). For example, the elegant work of Gong's group showcases the generality of this strategy, allowing arylation, alkylation and allylation of tertiary alkyl halides through Ni-catalyzed reductive cross-electrophile couplings[21–24]. In addition to the above mentioned direct coupling manifolds, car-bofunctionalization of 1,1-disubstituted or trisubstituted alkenes is gaining momentum (Fig. 1c)[28–32]. Representative examples in this vein include Shenvi's hydroarylation/alkylation of unac-tivated alkenes through Fe/Ni or Mn/Ni co-catalysis[29,30] and Brown's diarylation and arylboration of trisubstituted alkenes[31,32]. Although these methods enable access to *tert*-C−C linkages, development of strategically different approaches remain in high demand.

By exploiting iterative hydrometallation and β-hydride elim-ination, chain-walking enables the site-selective cross-coupling at positions remote to the initial metallation site[33,34]. Owing to the efforts of Sigman[35–37], Marek[38–40], Mazet[41–43], Martín[44–47], Zhu[48–57], and others[58–66], a collection of remote functionalizations, including arylation, alkylation, carboxylation, amination, borylation, and thiolation of unactivated alkenes or alkyl halides have been developed. Very recently, our team leveraged the fluorine-effect for a remote fluoro-alkenylation of unactivated alkyl bromides (Fig. 1d)[67]. Our system is like other remote functionalization reactions, where the driving force for chain-walking is moving the system lower on the energy landscape by positioning the metal center at a stabilizing position (usually limited to benzylic or alpha to boron). To expand the scope of remote functionalization reactions, alternative sites must be tar-geted, such as tertiary centers. With our continuing interest in remote functionalization, we have uncovered a mechanistically distinct and highly regioselective migratory 3,3-difluoroallyla-tion[68–71] of unactivated alkyl bromides at tertiary carbon centers. This undirected *tert*-C(sp³)−H functionalization nicely comple-ments existing methods for all-carbon quaternary center con-struction, especially when tertiary alkyl halide/metal reagents are not readily available or not stable. Notably, during the preparation of this manuscript, Zhu and co-workers reported a relevant Ni–H-catalyzed migratory defluorinative olefin cross-coupling[57].

Herein, we demonstrate that unactivated primary and sec-ondary alkyl bromides are competent precursors for generating tertiary alkyl coupling partners via Ni–H-mediated chain-walking (Fig. 1e).

## Results
**Reaction optimization**. A selection of alkyl halides was employed to react with α-trifluoromethylstyrene **2a**. After initial screening,

(bromomethyl)cyclohexane was successfully coupled with **2a** at the *tertiary* position with good regioselectivity (>20:1) 64% assay yield in the presence of NiBr₂·glyme, 6,6′-dimethyl-2,2′-bipyr-idine (**L1**) and Mn as terminal reductant (Table 1, entry 1, AY determined by integration of the ¹⁹F NMR spectrum against an internal standard). Given the pivotal role of ligands in Ni-catalyzed remote functionalizations, a series of bidentate *N*-donor ligands were examined to improve the reaction outcome. Sub-stitution next to the nitrogens of the bipy ligands was found indispensable. Without either one or two methyl groups posi-tioned ortho to the nitrogens, no product was observed. This observation is in accordance with previous reports (Supplemen-tary Table 1)[46,53,54]. We hypothesized that increasing the steric bulk around the metal coordination site would enhance the reaction efficiency. Thus, a series of increasingly bulky sub-stituents, such as ethyl, propyl, and butyl, were subsequently examined (entries 2–7). This study led to 6,6′-diethyl-2,2′-bipyridine (**L2**) as the top candidate, furnishing product **3a** in improved yield and comparable regioselectivity (72% AY and >20:1 regioisomeric ratio, Table 1, entry 2). In addition, with pyrox or terpyridine ligands, essentially no reaction occurred (Table 1, entries 8 and 9). A solvent screen revealed that THF was the optimal choice (Table 1, entries 10–12), allowing the forma-tion of the product in 89% isolated yield with >20:1 regioisomeric ratio. The influence of reductant was also examined. Mn proved superior to Zn, B₂pin₂, diethoxymethylsilane and HCOONa, which are commonly employed in reductive cross-coupling reactions (Supplementary Table 4).

**3,3-Difluoroallylation of unactivated alkyl bromides**. With the optimized reaction conditions in hand, the reaction scope of alkyl bromides was examined (Table 2). We found that a broad range of unactivated alkyl bromides were suitable substrates for the difluoroallylation. Cyclic alkyl bromides containing heteroatoms, such as oxygen and *N*-Boc, were well-tolerated, affording the 3,3-difluoroallylated products in 85 and 65% yields with excellent regioselectivities (**3b** and **3c**). Notably, a cyclic acetal was toler-ated to afford the desired product with good regioisomeric ratio, albeit in diminished yield (**3d**). Examination of the 5- and 7-membered carbocycles resulted in good regioselectivities (>11:1) with yields of 56 (**3e**) and 51% (**3f**) under the standard conditions. The lowered rr of 11:1 for **3e** may be due to increased strain in the β-H elimination transition state. We were pleased to find that acyclic alkyl bromides provided products containing quaternary centers in 63–65% and high regioselectivities (>20:1, **3g**–**3i**). Ester, ether, silyl ether, and phthalimide moieties were well tol-erated, affording the corresponding 3,3-difluoroallaion products in good yields (**3j**–**3m**). Interestingly, substrates containing two contiguous tertiary carbon centers only led to the migratory product at the proximal site (**3n**). Importantly, it was found that the migration could proceed over more than one C–C bond, albeit with progressively decreased reaction efficiency and regioisomeric ratio (**3o**, 47% yield with 7:1 rr and **3p**, 31% yield with 2:1 rr). Nonetheless, these results highlight the selectivity of the present catalytic system toward tertiary carbon centers over secondary and primary positions. This trend is also observed in the formation of products **3q** and **3r**, where *sec*-alkyl bromides reacted ultimately giving predominantly coupling products at the tertiary site. The intramolecular competition revealed that the tertiary carbon was more favorable than 1° or 2° and even pre-ferred over benzylic positions (**3s**). These findings stand in con-trast to previous disclosures[54]. To further distinguish reactivity between secondary and primary sites, *n*-propyl and *n*-butyl bromide were examined (**1t** and **1u**). It was found that coupling occurred more readily at the more congested secondary position

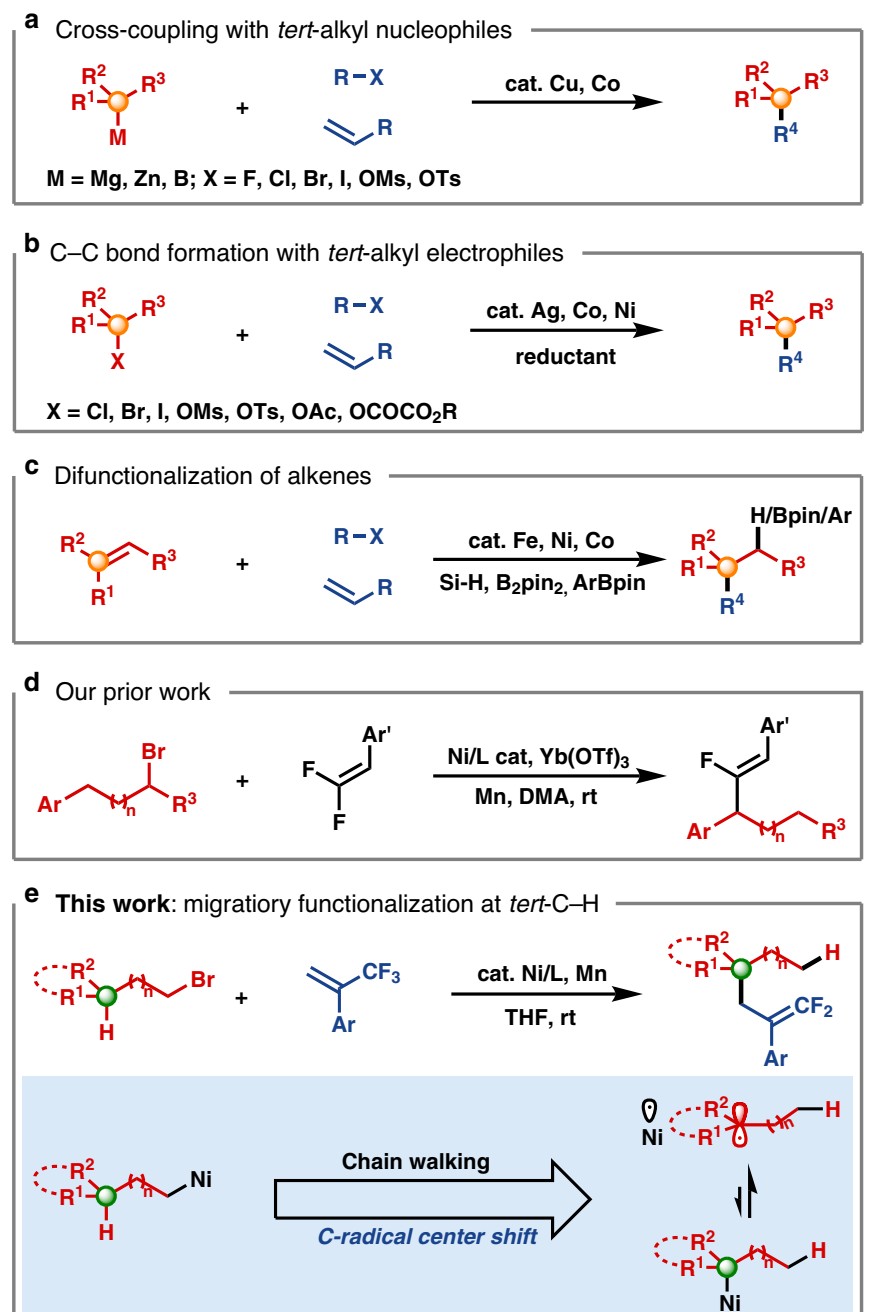

**Fig. 1 Transition metal catalyzed formation of quaternary centers. a** Standard cross-coupling approach, **b** cross-electrophile coupling, **c** difunctionalization of alkenes, **d** our prior work, **e** difluoroallylation of alkyl bromides with chain walking via transition-metal-catalyzed *tert*-C–C bond-formation.

(**3t**, 53% and rr >20:1; **3u**, 28%, rr >20:1). It is noteworthy that chain-walking to more congested positions, in the absence of stabilizing groups, has not been previously realized.

It is interesting that the present reaction system can also be used to functionalize remote benzylic positions with high efficiency and selectivity (**3v**, 90% yield, >20:1 rr and **3w**, 50% yield with 7:1 rr). Pleasingly, drug derived substrates performed well in the reaction (**3x** and **3y**), demonstrating the synthetic potential of the difluoroallylation in late-stage modification of complex molecules. Furthermore, the diastereoselectivity of this transformation was assessed with enantioenriched substrate **1z**, which delivered the migratory product **3z** with 5:1 dr. Not unexpectedly, the optimized reaction conditions were applicable

to the difluoroallylation of tertiary alkylbromide (**3g** from *tert*-BuBr). Finally, 1 mmol scale reaction was accomplished by using commercially available ligand (**L1**) with comparable efficiency, affording **3a** in 65% yield with >20:1 rr.

**Reaction scope with trifluoromethylalkenes**. We next evaluated different trifluoromethylalkene substrates in this transformation (Table 3). Reactions carried out with α-trifluoromethylstyrenes bearing a wide range of functional groups on the aryl moiety, such as ester, ketone, cyanide, $CF_3$, $OCF_3$, sulfone, Me and OMe, all underwent coupling smoothly to afford the desired products in good yields (47–81%) and excellent selectivities (all > 20:1, **3aa**–**3ai**). In addition, substrates containing Cl or F on the aryl were

**Table 1 Optimization of the reaction conditions[a].**

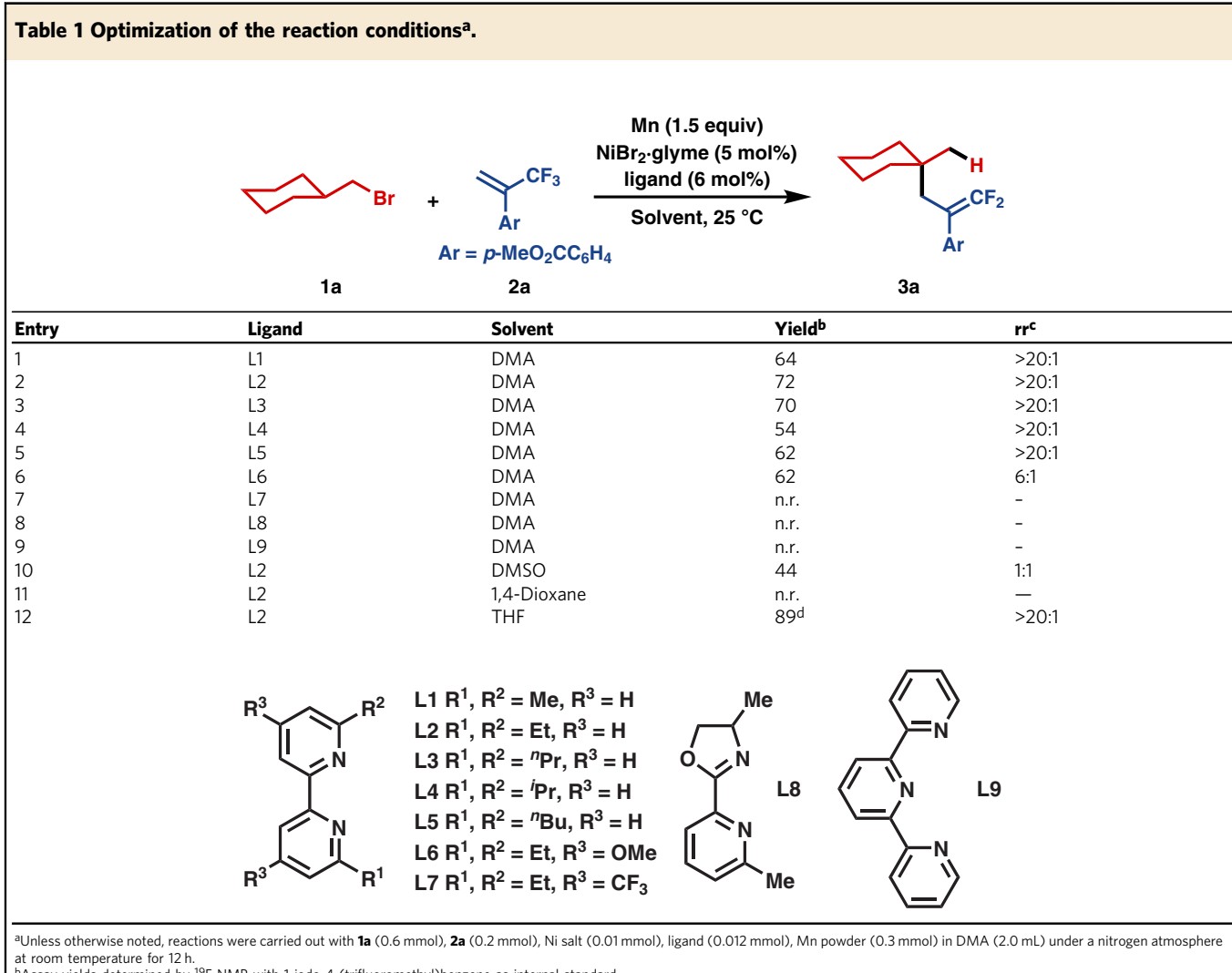

| Entry | Ligand | Solvent | Yield[b] | rr[c] |
|---|---|---|---|---|
| 1 | L1 | DMA | 64 | >20:1 |
| 2 | L2 | DMA | 72 | >20:1 |
| 3 | L3 | DMA | 70 | >20:1 |
| 4 | L4 | DMA | 54 | >20:1 |
| 5 | L5 | DMA | 62 | >20:1 |
| 6 | L6 | DMA | 62 | 6:1 |
| 7 | L7 | DMA | n.r. | – |
| 8 | L8 | DMA | n.r. | – |
| 9 | L9 | DMA | n.r. | – |
| 10 | L2 | DMSO | 44 | 1:1 |
| 11 | L2 | 1,4-Dioxane | n.r. | — |
| 12 | L2 | THF | 89[d] | >20:1 |

L1 $R^1$, $R^2$ = Me, $R^3$ = H
L2 $R^1$, $R^2$ = Et, $R^3$ = H
L3 $R^1$, $R^2$ = $^nPr$, $R^3$ = H
L4 $R^1$, $R^2$ = $^iPr$, $R^3$ = H
L5 $R^1$, $R^2$ = $^nBu$, $R^3$ = H
L6 $R^1$, $R^2$ = Et, $R^3$ = OMe
L7 $R^1$, $R^2$ = Et, $R^3$ = $CF_3$

[a]Unless otherwise noted, reactions were carried out with **1a** (0.6 mmol), **2a** (0.2 mmol), Ni salt (0.01 mmol), ligand (0.012 mmol), Mn powder (0.3 mmol) in DMA (2.0 mL) under a nitrogen atmosphere at room temperature for 12 h.
[b]Assay yields determined by $^{19}F$ NMR with 1-iodo-4-(trifluoromethyl)benzene as internal standard.
[c]Rr refers to the ratio of desired product to the sum of all the other regioisomers, as determined by the $^{19}F$ NMR analysis of the crude products.
[d]Isolated yields.

compatible with the transformation (**3aj–3al**, 51–78% yield, all >20:1 rr). Fortunately, heterocyclic trifluoromethyl alkene substrates (**3am-3ap**) were also well tolerated (45–80% yield, >20:1 rr).

**Reaction scope with other activated olefins**. To expand the scope of this transformation beyond trifluoromethylalkenes, other electron deficient olefins were examined. To our delight, acrylate, vinyl ketone, acrylonitrile, vinyl sulfone, and vinyl phosphonate derivatives were amenable under slightly modified reaction conditions. These substrates furnished migratory alkylation products in synthetically useful yields with excellent regioselectivities. These outcomes expand the synthetic reach of this *tert*-carbon-selective remote functionalization strategy, enabling the construction of quaternary carbon centers decorated with diverse functionality (Table 4).

To probe the mechanism of this migratory defluorinative allylation reaction, a set of control experiments were performed. To determine if chain-walking was indeed operating in the present system, the isotope-labelled substrate **1j-D** was examined (Fig. 2a). As expected, the deuterium located at the tertiary carbon was selectively transferred to the primary position. This result strongly supports the involvement of chain-walking. It is notable that no deuterium was observed at other positions in the product. We

hypothesized that radical intermediates may be involved and, therefore, conducted the reaction in the presence of TEMPO. The radical scavenger TEMPO suppressed the reaction and 97% of **2a** remained, supporting the involvement of radical intermediates (Fig. 2b, eq 1). In addition, when **2a** was replaced by allylic sulfone **6**, allylation proceeded, suggesting the existence of 3 °C-radical intermediate under the catalytic conditions (Fig. 2b, eq 2). To further elucidate the mode of C–C bond formation, cyclic β-pinene-derivative (**8**) was subjected to the reaction (Fig. 2c). The observation of radical intermediates would be expected to result in ring-opened products, whereas a two-electron process would leave the ring intact. In the event, the resulting ring-opening product (**9**) was exclusively obtained. To explain the results in Fig. 2, we propose a tertiary carbon radical is generated and participates in the crucial C–C bond formation step[72–75]. The oxidative addition of alkyl bromides to low-valent Ni catalysts usually takes place through a cascade of single electron transfer and alkyl radical generating steps[76]. Such transformations, therefore, can be viewed as unusual radical center shifts that are mediated by transition metal catalysts. We believe the steric hindrance encountered at tert-C–Ni linkage is conducive to the homolytic rupture of the C–Ni bond, affording tertiary carbon-centered radicals that are a sufficiently long lived to escape the solvent cage and selectively react with trifluoromethylalkene derivatives.

**Table 2 Scope with respect to alkyl bromides[a].**

3b 85% rr > 20:1

3c 65% rr > 20:1

3d 33% rr > 20:1

3e 56% rr = 11:1

3f 51% rr > 20:1

3g 65% rr > 20:1

3h 64% rr > 20:1

3i 63% rr > 20:1

3j 67% rr > 20:1

3k 74% rr > 20:1

3l 72% rr > 20:1

3m 52% rr > 20:1

3n 72% rr > 20:1

3o 47% rr = 7:1

3p 31% rr = 2:1

3q 63% rr = 9:1

3r 49% rr = 2:1

3s 55% rr = 12:1

3t 53% rr > 20:1

3u 28% rr > 20:1

3v 90% rr > 20:1

3w 50% rr = 7:1

*From isoxepac*
3x 70% rr > 20:1

*From indomethacin*
3y 65% rr >20:1

3z 55% rr > 20:1 dr = 5:1

3g 84%[b]

[a]See the Supplementary Information, pages 27–37, for experimental details. Rr refers to the ratio of desired product to the sum of all the other regioisomers, which was determined by the $^{19}$F NMR or GC analysis of the crude products.
[b]From *tert*-butyl bromide.

**Table 3 Scope with trifluoromethyl alkenes[a].**

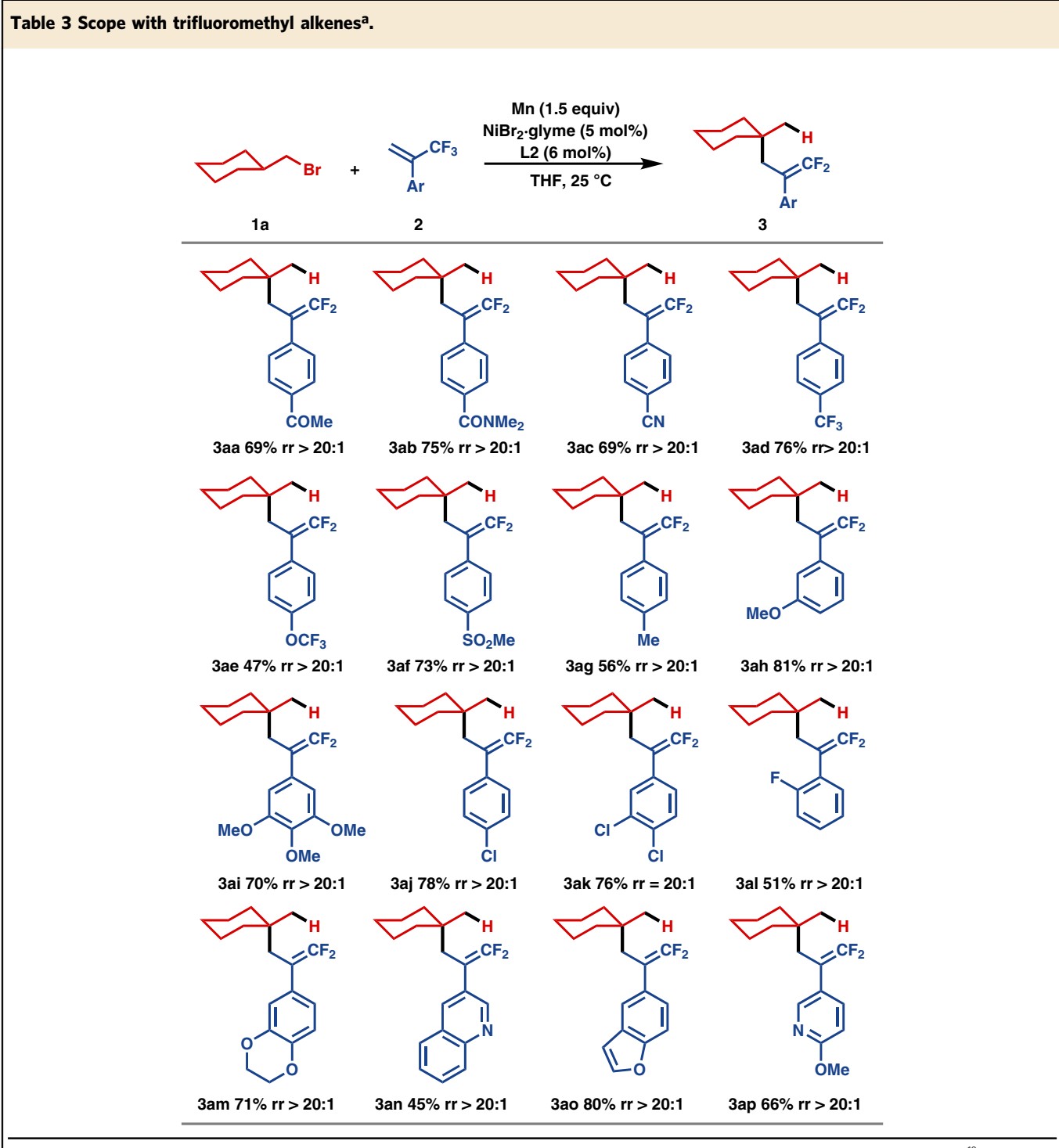

[a]See Supplementary Information, pages 37–44, for experimental details. Rr refers to the ratio of desired product to the sum of all the other regioisomers, which was determined by the [19]F NMR or GC analysis of the crude products.

Taken together, these findings demonstrate that a radical-engaged chain-walking manifold is involved, which accounts for the unusual selectivity that leads to functionalization at the more congested tertiary position. A proposed mechanism is outlined in Fig. 3. The reaction is initiated by the oxidative addition of alkyl bromide **1** to active Ni complex **I** to afford intermediate **II**. Subsequently, chain-walking of the nickel catalyst from the terminal carbon to the tertiary center via β-hydride elimination and insertion steps allows for the facile generation of *tert*-C–Ni complex **IV**. The Ni–C to the tertiary carbon bond has the lowest BDE and undergoes homolysis, generating a tertiary C-radical **V**. The radical can undergo addition to the trifluoromethylalkene **2** to form a new radical species **VI**. The newly-formed radical intermediate then recombines with the nickel complex to give rise to intermediate **VII**, which undergoes β-fluoride elimination to produce the observed product **3**, accompanied by the generation of F–Ni-complex **VIII**. Finally, reduction of Ni-complex **VIII** with Mn° closes the catalytic cycle by regenerating the active catalyst **I**. Whereas the radical manifold is consistent with the control experiments, the possible engagement of a 2-electron

**Table 4 Scope of activated alkenes[a].**

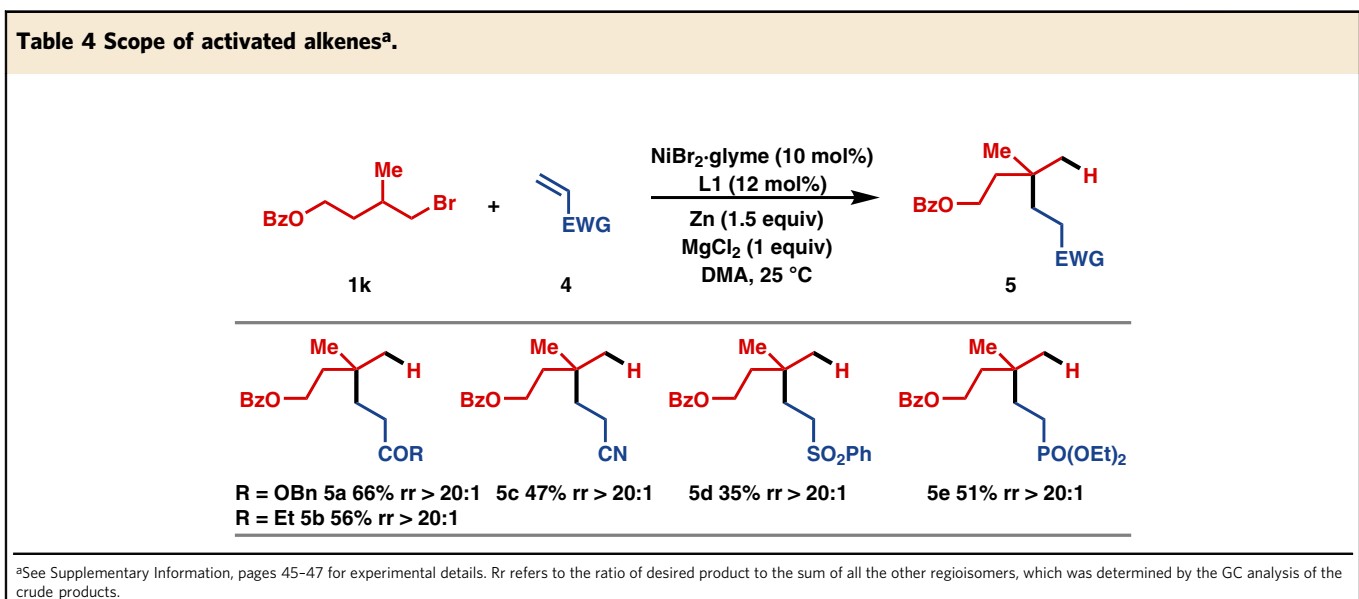

[a]See Supplementary Information, pages 45–47 for experimental details. Rr refers to the ratio of desired product to the sum of all the other regioisomers, which was determined by the GC analysis of the crude products.

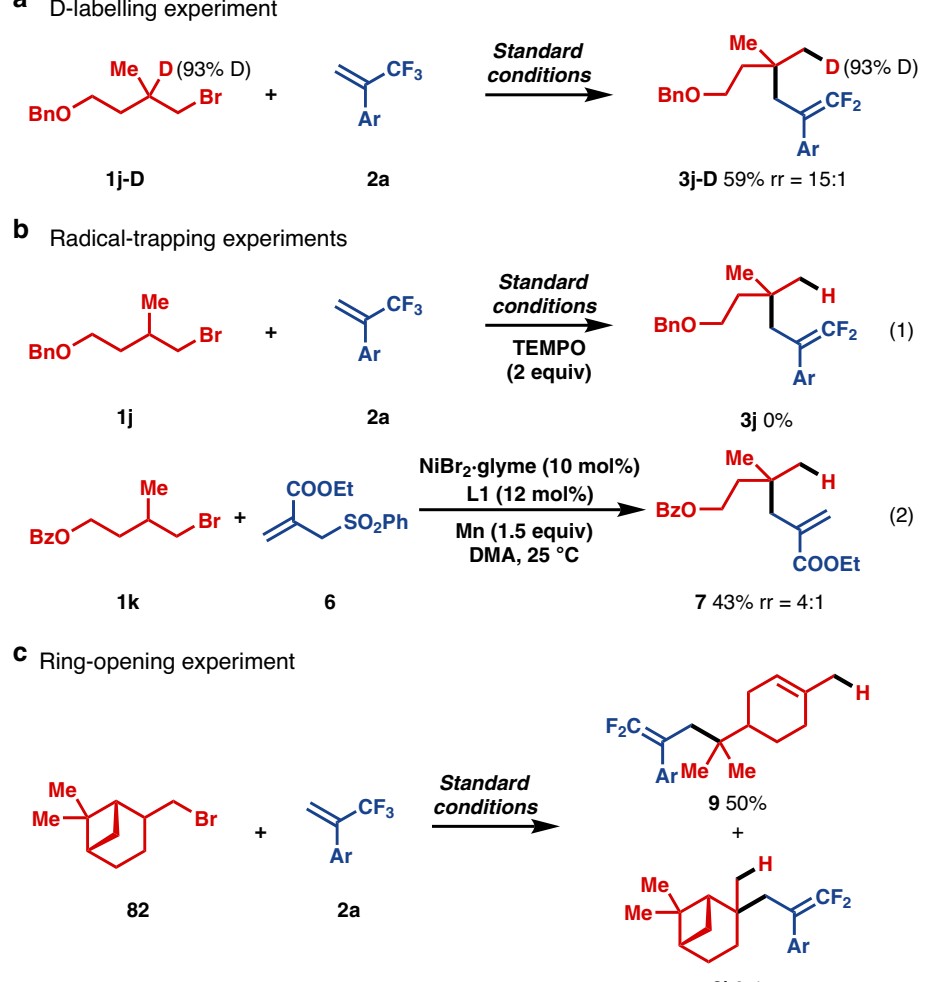

**Fig. 2 Control experiments (Ar = 4-MeO₂C-C₆H₄). a** A deuterium-labelling experiment; **b** radical trapping experiments; **c** ring-opening experiment.

**Fig. 3 Plausible mechanism.** Ni-catalyzed chain-walking process enables a C-radical translocation event, thus allowing migratory functionalization at the sterically more hindered position.

pathway relying on the direct addition of alkyl-Ni **IV** across the C–C double bond of **2**, cannot be ruled out.

## Discussion

In summary, a Ni-catalyzed reductive coupling for the synthesis of 1,1-difluoroalkenes containing quaternary centers is introduced. Difluoroalkenes are known bioisosteres for carbonyl groups in medicinal chemistry. The key to the success of this method is development of a Ni−H initiated remote functionalization of alkyl bromides. This approach enables a defluorinative 3,3-difluoroallylation of unactivated alkyl bromide substrates at sterically congested tertiary positions as an approach to selectively construct all-carbon quaternary centers. It is noteworthy that this transformation represents a rare case of C-radical transposition, which is enabled by a Ni−H chain-walking manifold. The successful development of this protocol demonstrates that readily available primary and secondary alkyl bromides can be used as progenitors for the construction of quaternary carbon-containing frameworks. In view of the potential impact of this strategy for remote functionalization, efforts to develop additional transformations are underway in these laboratories.

## Methods

**General procedure for the 3,3-difluoroallylation of unactivated alkylbromides.** To an oven-dried Schlenk tube equipped with a magnetic stir bar was added NiBr$_2$·glyme (3.1 mg, 0.01 mmol, 5.0 mol%), L2 (2.5 mg, 0.012 mmol, 6.0 mol%), Mn powder (16.5 mg, 0.3 mmol, 1.5 equiv). After the Schlenk tube was evacuated and filled with nitrogen for three cycles, THF (1.0 mL), compound **1** (0.6 mmol, 3.0 equiv) and compound **2** (0.2 mmol, 1.0 equiv) were added under nitrogen atmosphere. The Schlenk tube was maintained at 25 °C for 12 to 24 h. The reaction mixture was then diluted with ethyl acetate (10 mL) and washed with H$_2$O (10 mL). The aqueous layer was extracted with ethyl acetate (10 mL × 2). The combined

organic layers were washed with water (10 mL), brine (10 mL) and dried over Na$_2$SO$_4$. After solvent was removed under reduced pressure, the crude residue was analyzed by $^{19}$F NMR with 1-iodo-4-(trifluoromethyl)benzene as internal standard, and then the mixture was purified by column chromatography or preparative TLC on silica gel to afford the desired product.

## Data availability

The authors declare that all the data supporting the findings of this study are available within the paper and its Supplementary Information files, or from the corresponding authors upon request.

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

## Acknowledgements

We gratefully acknowledge the financial support of the National Natural Science Foundation of China (21801131), the Natural Science Foundation of Jiangsu Province (BK20170992), the "Thousand Talents Plan" Youth Program, and the "Jiangsu Specially-Appointed Professor Plan". P.J.W. thanks the US National Science Foundation (CHE-1902509).

## Author contributions

C.Z. and C.F. conceived and directed the project. Z-Y.L. performed the experiments. L.T., H.Z., and Y-F.Z. prepared some of the substrates. C.Z. and C.F. analyzed the results and wrote the paper. P.J.W. discussed the chemistry, suggested experiments and revised the paper.

## Competing interests

The authors declare no competing interests.
