## [Peer Review File · Nature Communications]

Reviewers' Comments:

Reviewer #1:

Remarks to the Author:

The authors describe a method for the formation of quaternary centers utilizing Ni-catalyzed chain walking of primary and secondary alkyl bromides. The scope investigates cyclic and acyclic aliphatic components and a range of aryl group functionalities are tolerated. In terms of method development, the recently published Angew paper from Zhu (ref 71) most significantly hinders the novelty of this transformation, albeit utilizes different starting materials. Additionally, as the authors point out, the Zhu paper lacks mechanistic inquires for the proposed intermediates. The current manuscript does a better investigation to support the proposed mechanism. The low BDE of the tert-alkyl Ni species facilitating a persistent radical effect that ultimately allows for a free radical addition into the vinyl coupling partner seems plausible to me. Overall, the method offers a nice installment into the repertoire of Ni-catalyzed chain walking that demonstrates a means for obtaining tertiary selectivity over primary, secondary, and benzylic functionalization.

I have several questions pertaining to the scope.

1) The authors give no explanation for some of the lower regioselectivity. For example, the 5-membered ring vs 6-membered ring. Conceivably a consequence of difference in orbital overlap and positioning of the beta-hydrogens to facilitate the initial beta-hydride elimination? Would be better to give scientific rationale/hypothesis to better guide the reader and curious mind. The authors are significantly the experts on this subject in comparison to most of the readers. I would assume the authors have thought about many of these difference in regioselectivity. It would be instructive to elaborate.

2) The case of 3k makes me curious about the regioselectivity when extended beyond an additional methylene. Does the method lose all regioselectivity? With the high selectivity observed for tertiary product formation, it is surprising to me that the chain walking would be hampered by the additional methylene. Can the authors please comment on why they think this occurs and also how common this is with regards to other Ni-catalyzed chain walking. I was under the impression that, at least in the case of benzylic functionalization, the chain walking was selective irrespective of distance. Since this method is more selective for tertiary than even benzylic, this length dependence is odd to me.

3) For 3o, what is the residual mass balance? Surprising that 3n is approximately twice the yield.

4) The scope is predominantly carbon based. Is the functional group tolerance poor due to synthesis of the alkyl bromides or is the chain walking highly sensitive to proximal functional groups? I realize 3b and 3c have non-carbon atoms, but the scope is still predominantly carbon.

In my experience, some Ni(II) salts can have a low solubility in 1,4-dioxane. Did the authors try preforming the NiLn complex and then subjecting the complex to the reaction? Seems deceptive to say that there is n.r. in 1,4-dioxane if it's just a consequence of metal solubility, especially given that the optimized reaction is in THF. For similar Ni complexes, one can stir Nibr2glyme and the Ln in THF under argon overnight and then filter the ppt or rotovap.

I recommend for publication after revisions.

Reviewer #2:

Remarks to the Author:

Feng and Walsh report in their manuscript Ni-catalyzed migratory difluoroallylation of simple alkyl bromides. Products with a fully substituted carbon center are accessed from the substrates of CF₃-substituted vinyl arenes. Transition metal catalyzed reductive coupling reactions have been reported in many systems recently, including nickel-catalyzed reactions using the same CF₃-substituted vinyl arenes (e.g., references 66 and 68 in this manuscript, Angew. Chem. Int. Ed.

2020, 59, 5398, etc.). In my opinion, this work is conceptually related to those work. The novelty may only lie in the involvement of radical transposition to a tertiary one. I don't support acceptance of this work for publication in its current form. Only if the following major revisions are properly addressed, it may be suitable for publication in a top journal such as Nat. Commun.

a) To demonstrate the potential for generating all-carbon quaternary centers using this new migratory strategy, and to differentiate it from the previous work, the authors should try to extend this chemistry to substrates other than α -trifluoromethylstyrenes.

b) For the deuterium-labeling experiment, the authors should determine the ratios of deuterium both in the substrate and product.

c) As a tertiary carbon radical was proposed in the reaction, the authors should perform some trapping experiments to provide support for it.

Reviewer #3:

Remarks to the Author:

In this paper, Patrick J. Walsh, Chao Feng and co-workers report an interesting remote functionalization using a Ni catalyzed migratory 3,3-difluoroallylation of unactivated alkyl bromides which allows the formation of compounds bearing a quaternary centre starting from easily available starting materials.

Overall, this is a sound piece of work in a very competitive area of research, which I believe will be appealing to the readership of Nature Communications; the concept is interesting, the method is relatively efficient (broad scope with moderate to high yields), the mechanism proposed is reasonable and supported by various experiments, the manuscript presents well, the procedures are clearly reported in the supplementary material section and the references are appropriate and correct. For all these reasons, I would be inclined to support its publication, however some major revisions are needed.

- The closely related work by Rueping and co-workers [Angew. Chem. Int. Ed. 2020 (DOI: 10.1002/anie.201915418)], that of Zhu and co-workers [Angew. Chem. Int. Ed. 2020 (DOI: 10.1002/anie.202001742) and Nat. Commun. 2019, 10, 1752], and perhaps that of Ichakawa (ACS Catal. 2015, 5, 5947), should be cited in the introduction. Most importantly, the NiH-catalyzed migratory defluorinative olefin cross-coupling developed by Zhu and co-workers (ref 71 in the manuscript) should appear in the introduction rather than in the discussion of the mechanism.
- The screening of the reaction conditions is comprehensively presented, however some information should be added: (1) were there any other terminal reductants tested other than Mn? If so, those should be added, if not, perhaps some should be tested, (2) Were there any other solvents tested other than DMA, NMP, DMSO, THF and 1,4-dioxane, in any case, could the authors comment on the influence of the solvent? (3) Unless there is a typo in Table S2 entry 4 and Table S3, entry 3 in the SI, there is an issue with the quantification method using ^{19}F NMR and 1-iodo-4-(trifluoromethyl)benzene as an internal standard (NMR yield indicates 77% while the isolated yield is 89%). The reactions are typically run on a 0.2 mmol scale of the trifluoromethyl alkene. How much internal standard is added to reaction mixture? This information should be added in the general procedure in the SI along with a representative NMR.
- The authors indicate "Herein, we demonstrate that unactivated primary and secondary alkyl bromides are competent precursors [...]" however only one example of a secondary alkyl bromide is shown (compound 3l) and not only that, it comes from a symmetrical precursor. The authors should definitely test the limits of their method by selecting substrates which could potentially lead to a different outcome and also allow them to tame their reaction. This is definitely more interesting than changing the gem-dimethyl (compound 3f) by a gem-ethyl/butyl (compound 3g).

One can wonder what the outcome would be if the isopropyl was replaced by a phenyl in a compound such as 1l for example. Also, even if it is out of scope, it is questionable whether a tertiary alkyl bromide would work. Another important aspect concerns the stereoselectivity of the process; would the reaction be diastereoselective if one of the substituents was chiral? Have the authors envisioned an enantioselective process using a chiral ligand? I think these aspects deserve to be included even if the results are not as appealing as one would want.

- While the scope in terms of the trifluoromethyl alkene is sufficient (at least to my opinion), that of the alkyl bromide is a bit limited with some examples that are clearly redundant. Like I mentioned in one of my previous comments, I'd rather see more key examples than just simple alkyl chain alteration (3d vs 3e, 3f vs 3g vs 3h...). Most importantly in the context of this work, I would like to see how far this C-radical transposition could go. In other words, I would like to see more examples like 3k (e.g. n = 2, 5 and 10).
- Even though the reaction itself is very appealing, the synthetic utility of the gem-difluoroalkenes should also be showcased through various post-functionalizations. At the very least, several relevant references relative to the properties of lightly fluorinated moieties or simply difluoroalkene should be added and mentioned in the manuscript.
- The authors should at least run one reaction on a 1 mmol scale.
- The authors conducted several experiments to ascertain the radical pathway. All three, in addition to several examples in the scope, were well-chosen and bring a solid rationale to the proposed radical pathway, however, as mentioned by the authors, the heterolytic cleavage of the C-Ni bond can't be ruled out (reference 70). Perhaps this could be included in the mechanism in Scheme 2 using a lighter colour and discussed in the main text.
- The SI clearly needs to be improved:
 1. A description of the ¹H NMR and ¹³C NMR spectra of all the compounds that have been previously reported in the literature should be provided (e.g. S1-7, 1e, 1i, 1j, 1q...).
 2. 1l should be fully characterized (¹H and ¹³C NMR, HRMS...).
 3. The synthesis of compounds 2a-2q should be included (general procedures at least). The yields and NMR data are missing.
 4. The synthesis of compounds L2-L8 should be included (general procedures at least). The yields and NMR data are missing.
 5. The optimisation should include a representative yield determination using ¹⁹F NMR and the chosen internal standard (with spectrum, integration and calculation method).
 6. Table S3: The rr for entry 2 is missing.
 7. The amount of product 3a-3af isolated should be added in addition to the yield.
 8. All ¹⁹F NMR spectra of the purified final products should include a zoom of the region containing the signals.
 9. Most importantly, the rr determinations on the crude reaction mixtures are not accurate (the signal at -88 ppm seems to be relevant but that "impurity/by-product" has not been characterized nor integrated). Several ¹⁹F NMRs show some remaining impurities after purification.
- The reaction using TEMPO did not give the title product which supports the radical mechanism,

however, what was the outcome of that reaction?

- On a minor note, Table 2, compound 3g, the C–H bond should be in black.

Reviewer #1:

The authors describe a method for the formation of quaternary centers utilizing Ni-catalyzed chain walking of primary and secondary alkyl bromides. The scope investigates cyclic and acyclic aliphatic components and a range of aryl group functionalities are tolerated. In terms of method development, the recently published Angew paper from Zhu (ref 71) most significantly hinders the novelty of this transformation, albeit utilizes different starting materials. Additionally, as the authors point out, the Zhu paper lacks mechanistic inquires for the proposed intermediates. The current manuscript does a better investigation to support the proposed mechanism. The low BDE of the tert-alkyl Ni species facilitating a persistent radical effect that ultimately allows for a free radical addition into the vinyl coupling partner seems plausible to me. Overall, the method offers a nice installment into the repertoire of Ni-catalyzed chain walking that demonstrates a means for obtaining tertiary selectivity over primary, secondary, and benzylic functionalization.

I have several questions pertaining to the scope.

Q: 1) The authors give no explanation for some of the lower regioselectivity. For example, the 5-membered ring vs 6-membered ring. Conceivably a consequence of difference in orbital overlap and positioning of the beta-hydrogens to facilitate the initial beta-hydride elimination? Would be better to give scientific rationale/hypothesis to better guide the reader and curious mind. The authors are significantly the experts on this subject in comparison to most of the readers. I would assume the authors have thought about many of these difference in regioselectivity. It would be instructive to elaborate.

A: Indeed, we hypothesize that the lower rr for 5-membered ring derived substrate is likely due to the higher barrier in the crucial β -H elimination step. The five-membered ring suffers more sever ring-strain in the transition state to beta-hydrogen elimination. Such explanation has been added to the main text as “The lowered rr of 11:1 for **3e** may be due to increased strain in the β -H elimination transition state.”

Q: 2) The case of **3k** makes me curious about the regioselectivity when extended beyond an additional methylene. Does the method lose all regioselectivity? With the high selectivity observed for tertiary product formation, it is surprising to me that the chain walking would be hampered by the additional methylene. Can the authors please comment on why they think this occurs and also how common this is with regards to other Ni-catalyzed chain walking. I was under the impression that, at least in the case of benzylic functionalization, the chain walking was selective irrespective of distance. Since this method is more selective for tertiary than even benzylic, this length dependence is odd to me.

A: We tested alkyl bromides with bromide located further from the tertiary carbon. It was found that extension of the carbon chain led to progressively decreased regioselectivity. It is more difficult to differentiate between secondary and tertiary radicals than it is between alkyl vs. aryl, as in other systems. In our system, competing insertion pathways are likely closer in energy to the desired pathway. The results of 1-bromo-3-methylbutane (**3o**, n = 1) and 1-bromo-4-methylpentane (**3p**, n = 2) have been added into the revised manuscript and supplementary information.

Q: 3) For **3o**, what is the residual mass balance? Surprising that **3n** is approximately twice the yield.

A: Because (alkene)[Ni]-H intermediates (see catalytic cycle in the manuscript) can presumably dissociate, side products involving hydrogenation or hydrodefluorination of the trifluoromethylalkene are observed. When **1n** was employed both the hydrogenated and hydrodefluorinated by-products were observed along with **3n** (**3t** in the revised manuscript). Likewise, the lower yield from *n*-butane **3o** (**3u** in the revised manuscript) may be due to generation of isomeric butenes, which are gasses.

Q: 4) The scope is predominantly carbon based. Is the functional group tolerance poor due to synthesis of the alkyl bromides or is the chain walking highly sensitive to proximal functional groups? I realize **3b** and **3c** have non-carbon atoms, but the scope is still predominantly carbon.

A: Additional substrates containing heteroatom proximal to the tertiary carbon (**3d**, **3l** and **3m** in the revised manuscript) were added to the revised manuscript and afforded moderate to good yields of desired products.

Q: In my experience, some Ni(II) salts can have a low solubility in 1,4-dioxane. Did the authors try preforming the NiL_n complex and then subjecting the complex to the reaction? Seems deceptive to say that there is n.r. in 1,4-dioxane if it's just a consequence of metal solubility, especially given that the optimized reaction is in THF. For similar Ni complexes, one can stir NiBr₂glyme and the L_n in THF under argon overnight and then filter the ppt or rotovap.

A: According to the referee's suggestion, the complex of NiBr₂·glyme and L2 was preformed by stirring in THF at room temperature for 8 h. Subsequently, the solvent was removed by evaporation in the glovebox. With the Ni-L2 complex the reaction was carried out in 1,4-dioxane and the desired product **3a** was not observed by crude ¹⁹F NMR.

For Reviewer #2 (Remarks to the Author):

Feng and Walsh report in their manuscript Ni-catalyzed migratory difluoroallylation of simple alkyl bromides. Products with a fully substituted carbon center are accessed from the substrates of CF₃-substituted vinyl arenes. Transition metal catalyzed reductive coupling reactions have been reported in many systems recently, including nickel-catalyzed reactions using the same CF₃-substituted vinyl arenes (e.g., references 66 and 68 in this manuscript, *Angew. Chem. Int. Ed.* 2020, 59, 5398, etc.). In my opinion, this work is conceptually related to those work. The novelty may only lie in the involvement of radical transposition to a tertiary one. I don't support acceptance of this work for publication in its current form. Only if the following major revisions are properly addressed, it may be suitable for publication in a top journal such as *Nat. Commun.*

Q: a) To demonstrate the potential for generating all-carbon quaternary centers using this new migratory strategy, and to differentiate it from the previous work, the authors should try to extend this chemistry to substrates other than α -trifluoromethylstyrenes.

A: The present migratory functionalization system was further investigated with activated olefins, including benzyl acrylate, ethyl vinyl ketone, acrylonitrile, vinyl phenyl sulfone, and vinyl phosphonate. We were pleased to find that alkylation at the tertiary carbon through a similar chain-walking proceeded under slightly modified reaction conditions. These exciting results further distinguish our study from Zhu's work. These results have been summarized in Table 4 in the revised manuscript.

Q: b) For the deuterium-labeling experiment, the authors should determine the ratios of deuterium both in the substrate and product.

A: The D-content was determined to be 93% for both alkyl bromide **1j-D** and the difluoroallylation product **3j-D** by ^1H NMR. This is described in the Scheme 1a.

Q: c) As a tertiary carbon radical was proposed in the reaction, the authors should perform some trapping experiments to provide support for it.

A: We performed the reaction with allyl sulfone instead of trifluoromethylalkene, whereby the tertiary carbon allylated product was isolated in 43% yield with 4:1 rr (Scheme 2b, eq 2 in the revised manuscript). This result lends further support for the *tert*-C radical based mechanism.

For Reviewer #3 (Remarks to the Author):

In this paper, Patrick J. Walsh, Chao Feng and co-workers report an interesting remote functionalization using a Ni catalyzed migratory 3,3-difluoroallylation of unactivated alkyl bromides which allows the formation of compounds bearing a quaternary centre starting from easily available starting materials.

Overall, this is a sound piece of work in a very competitive area of research, which I believe will be appealing to the readership of Nature Communications; the concept is interesting, the method is relatively efficient (broad scope with moderate to high yields), the mechanism proposed is reasonable and supported by various experiments, the manuscript presents well, the procedures are clearly reported in the supplementary material section and the references are appropriate and correct. For all these reasons, I would be inclined to support its publication, however some major revisions are needed.

Q: • The closely related work by Rueping and co-workers [Angew. Chem. Int. Ed. 2020 (DOI: 10.1002/anie.201915418)], that of Zhu and co-workers [Angew. Chem. Int. Ed. 2020 (DOI: 10.1002/anie.202001742) and Nat. Commun. 2019, 10, 1752], and perhaps that of Ichakawa (ACS Catal. 2015, 5, 5947), should be cited in the introduction. Most importantly, the NiH-catalyzed migratory defluorinative olefin cross-coupling developed by Zhu and co-workers (ref 71 in the manuscript) should appear in the introduction rather than in the discussion of the mechanism.

A: The references have been cited properly according to the referee's suggestion. Also, the statement "Notably, during the preparation of this manuscript, Zhu and co-workers reported a relevant NiH-catalyzed migratory defluorinative olefin cross-coupling.⁵⁷" was added into the introduction of the revised manuscript.

Q: The screening of the reaction conditions is comprehensively presented, however some information should be added: (1) were there any other terminal reductants tested other than Mn? If so, those should be added, if not, perhaps some should be tested, (2) Were there any other solvents tested other than DMA, NMP, DMSO, THF and 1,4-dioxane, in any case, could the authors comment on the influence of the solvent? (3) Unless there is a typo in Table S2 entry 4 and Table S3, entry 3 in the SI, there is an issue with the quantification method using ^{19}F NMR and 1-iodo-4-(trifluoromethyl)benzene as an internal standard (NMR yield indicates 77% while the isolated yield is 89%). The reactions are typically run on a 0.2 mmol scale of the trifluoromethyl alkene. How much internal standard is added to reaction mixture? This information should be added in the general procedure in the SI along with a representative NMR.

A: Concerning the terminal reductant, Zn, HCOONa, B_2pin_2 and diethoxymethylsilane were examined instead of Mn but failed to afford improvement in yield (see Table S4). In accordance with the previous reports on NiH-catalyzed remote functionalization, polar solvents were demonstrated to be useful for the reaction with varied regioselectivity. Presumably, the difference in solubility of the reaction components and the ability of solvent to bind to Ni intermediates might account for the different reaction outcomes. The reactions conducted under optimal conditions were repeated twice for the determination of assay yields. Based on the two runs, assay yield and isolated yield were confirmed as 80% and 84%, respectively. To determine the assay yield, 0.133 mmol of 1-iodo-4-(trifluoromethyl)benzene was added to the crude reaction products as internal standard for the ^{19}F NMR analysis. A detailed method to calculate the assay yield has been added to the supplementary information using representative example (page S23).

Q: The authors indicate "Herein, we demonstrate that unactivated primary and secondary alkyl bromides are competent precursors [...]" however only one example of a secondary alkyl bromide is shown (compound 3l) and not only that, it comes from a symmetrical precursor. The authors should definitely test the limits of their method by selecting substrates which could potentially lead to a different outcome and also allow them to tame their reaction. This is definitely more interesting than changing the gem-dimethyl (compound 3f) by a gem-ethyl/butyl (compound 3g). One can wonder what the outcome would be if the isopropyl was replaced by a phenyl in a compound such as 1l for example. Also, even if it is out of scope, it is questionable whether a tertiary alkyl bromide would work. Another important aspect concerns the stereoselectivity of the process; would the reaction be diastereoselective if one of the substituent was chiral? Have the authors envisioned an enantioselective process using a chiral ligands? I think these aspects deserve to be included even if the results are not as appealing as one would want.

A: Secondary alkyl bromides are more challenging substrates to achieve good regioselectivity due to the smaller difference between the reactivity of 2° and 3° carbon centers. (1-bromo-2-methylpropyl)benzene failed to give any cross-coupling product with the trifluoromethylalkene remained (Scheme below).

A successful example is (1-bromoethyl)cyclohexane (**3r** in the revised manuscript) provided the desired product, albeit with 2:1 rr. Also, a chiral substrate methyl (1R,2R)-2-(bromomethyl)cyclohexane-1-carboxylate was examined which led to the desired product in 55% with > 20:1 rr and 5:1 dr (**3z** in the revised manuscript). The tertiary alkyl bromide afforded

the desired product in good yield (**3g** from *tert*-butyl bromide in the revised manuscript). The preliminary investigation of enantioselective version with chiral box or Pybox ligands (L12-L14) was conducted. Unfortunately, these ligands resulted in no desired product (Supplementary Information, page S24).

Q: While the scope in terms of the trifluoromethyl alkene is sufficient (at least to my opinion), that of the alkyl bromide is a bit limited with some examples that are clearly redundant. Like I mentioned in one of my previous comments, I'd rather see more key examples than just simple alkyl chain alteration (**3d** vs **3e**, **3f** vs **3g** vs **3h**...). Most importantly in the context of this work, I would like to see how far this C-radical transposition could go. In other words, I would like to see more examples like **3k** (e.g. $n = 2, 5$ and 10).

A: Alkyl bromides with longer carbon chains were investigated. It was found that the reaction efficiency and regioselectivity decreased with increasing the distance between the bromide and tertiary center. These results (**3g**, **3o** and **3p**) have been added in the revised manuscript. As noted above, the selectivity between secondary and tertiary is more difficult than between alkyl and benzylic in other systems. Longer distances between the bromide and the tertiary carbon result in an increase of side products (causing the lower rr).

Q: Even though the reaction itself is very appealing, the synthetic utility of the gem-difluoroalkenes should also be showcased through various post-functionalisations. At the very least, several relevant references relative to the properties of lightly fluorinated moieties or simply difluoroalkene should be added and mentioned in the manuscript.

A: Regarding the further application of this method, we added references into the main text which illustrate the pharmaceutical and synthetic utilities of gem-difluoroalkenes (ref. 69-71).

Q: The authors should at least run one reaction on a 1 mmol scale.

A: A 1 mmol scale reaction was performed with the commercially available ligand 6,6'-dimehtyl-2,2'-bipyridine and afforded the desired product **3a** in 65% isolated yield with >20:1 rr (see the Supplementary Information, page S43).

Q: The authors conducted several experiments to ascertain the radical pathway. All three, in addition to several examples in the scope, were well-chosen and bring a solid rationale to the proposed radical pathway, however, as mentioned by the authors, the heterolytic cleavage of the C-Ni bond can't be ruled out (reference 70). Perhaps this could be included in the mechanism in Scheme 2 using a lighter colour and discussed in the main text.

A: The possible 2-electron pathway has been added into the Scheme 2 and the relevant statement was added as "Whereas the radical manifold is consistent with the control experiments, the

possible engagement of a 2-electron pathway relying on the direct addition of alkyl-Ni **IV** across the C–C double bond of **2**, cannot be ruled out.”

Q: The SI clearly needs to be improved:

1. A description of the ¹H NMR and ¹³C NMR spectra of all the compounds that have been previously reported in the literature should be provided (e.g. S1-7, 1e, 1i, 1j, 1q...).
2. 11 should be fully characterized (¹H and ¹³C NMR, HRMS...).
3. The synthesis of compounds 2a-2q should be included (general procedures at least). The yields and NMR data are missing.
4. The synthesis of compounds L2-L8 should be included (general procedures at least). The yields and NMR data are missing.
5. The optimisation should include a representative yield determination using ¹⁹F NMR and the chosen internal standard (with spectrum, integration and calculation method).
6. Table S3: The rr for entry 2 is missing.
7. The amount of product 3a-3af isolated should be added in addition to the yield.
8. All ¹⁹F NMR spectra of the purified final products should include a zoom of the region containing the signals.
9. Most importantly, the rr determinations on the crude reaction mixtures are not accurate (the signal at -88 ppm seems to be relevant but that "impurity/by-product" has not been characterized nor integrated). Several ¹⁹F NMRs show some remaining impurities after purification.

A: The revisions of SI raised by the referee were addressed as below:

- 1-4. The spectroscopic data of alkyl bromides, trifluoroalkenes, ligands and relevant intermediates were added into the Supplementary Information.
5. The method to determine the NMR yield was described in the Supplementary Information.
6. The missing rr in entry 2 of Table S3 was added.
7. The amount of products were added accordingly in the Supplementary Information.
8. The expanded view of ¹⁹F NMR was added.
9. For the crude ¹⁹F NMR containing impurities overlapping the signal of regioisomers, GCMS analysis was employed to identify the rr. In the cases of compounds **3r**, **3w**, **3y**, **3ab**, and **3ac** (**3aa**, **3af**, **3ah**, **3ak** and **3al** in the revised manuscript), the GC charts were added to Supplementary Information instead of the corresponding crude ¹⁹F NMR spectra. Also, the ¹⁹F NMR of compounds **3d**, **3h**, **3n** and **3m** (**3e**, **3i**, **3t** and **3w** in the revised manuscript) were re-purified.

Q: • The reaction using TEMPO did not give the title product which supports the radical mechanism, however, what was the outcome of that reaction?

A: Addition of TEMPO shutdown the reaction completely, wherein the **2a** remained in 97% (data added to text). We explored the evidence for the radical mechanism further with allylic sulfone instead of trifluoromethylalkene. In the event, the tertiary C-radical was trapped through allylation, thus providing additional support for the proposed radical manifold (Scheme 2b, eq 2).

Q: • On a minor note, Table 2, compound 3g, the C–H bond should be in black.

A: The structure of **3g** (**3h** in the revised manuscript) was amended accordingly.

Reviewers' Comments:

Reviewer #1:

Remarks to the Author:

The authors have added significantly to their scope section to distinguish this methodology from analogous methods in the literature. I believe that their additions have adequately addressed my concerns and questions as well as those of the other reviewers. I appreciate the additional explanations pertaining to selectivity of the substrates as this will guide the future user and overall makes the work more scientific. I think the additional scope examples demonstrate Giese-like behavior and further support the hypothesis of a tertiary radical. I support publication of this manuscript.

Reviewer #2:

Remarks to the Author:

I have reviewed the original submission and I am happy to see that more extended substrate scope was indeed worked out by the authors. My other questions were also addressed nicely. I support publication of this work in Nat. Commun.

Reviewer #3:

Remarks to the Author:

Like I wrote in my earlier review, I think that this is a very nice piece of work which has gained additional impact after the comments made by the three referees were taken into account. The concept is very appealing, the method particularly effective, the mechanism proposed is reasonable and supported by various experiments, the manuscript presents well, the procedures are clearly reported in the supplementary material section and the missing references I had mentioned have been added. Every point raised by the referees have been taken into consideration. The authors made the effort to screen some additional conditions, to prepare various additional key products, and to re-purify all the products that contained some impurities.

Ultimately, this is a very nice paper that I believe will be appealing to the readership of Nature Communications.

I do have however have two very minor comments. The first one concerns Figure 1. Indeed, I find the representation of the group which is introduced (the blue sphere marked by a C) not super intuitive. Second, the general structure of product 5 in Table 4 is misleading. I would have something that relates to compound 1k. In other words, I would keep the common backbone in red.

Reviewer #1 (Remarks to the Author):

The authors have added significantly to their scope section to distinguish this methodology from analogous methods in the literature. I believe that their additions have adequately addressed my concerns and questions as well as those of the other reviewers. I appreciate the additional explanations pertaining to selectivity of the substrates as this will guide the future user and overall makes the work more scientific. I think the additional scope examples demonstrate Giese-like behavior and further support the hypothesis of a tertiary radical. I support publication of this manuscript.

Reviewer #2 (Remarks to the Author):

I have reviewed the original submission and I am happy to see that more extended substrate scope was indeed worked out by the authors. My other questions were also addressed nicely. I support publication of this work in Nat. Commun.

Reviewer #3 (Remarks to the Author):

Like I wrote in my earlier review, I think that this is a very nice piece of work which has gained additional impact after the comments made by the three referees were taken into account. The concept is very appealing, the method particularly effective, the mechanism proposed is reasonable and supported by various experiments, the manuscript presents well, the procedures are clearly reported in the supplementary material section and the missing references I had mentioned have been added. Every point raised by the referees have been taken into consideration. The authors made the effort to screen some additional conditions, to prepare various additional key products, and to re-purify all the products that contained some impurities. Ultimately, this is a very nice paper that I believe will be appealing to the readership of Nature Communications.

Q: I do have however have two very minor comments. The first one concerns Figure 1. Indeed, I find the representation of the group which is introduced (the blue sphere marked by a C) not super intuitive. Second, the general structure of product 5 in Table 4 is misleading. I would have something that relates to compound 1k. In other words, I would keep the common backbone in red.

A: The representation of incoming group in Figure 1a-c was changed to R⁴. The general structure of compound 5 in Table 4 was amended.